# Application of the Schroth Method in the Treatment of Idiopathic Scoliosis: A Systematic Review and Meta-Analysis

**DOI:** 10.3390/ijerph192416730

**Published:** 2022-12-13

**Authors:** Vanja Dimitrijević, Tijana Šćepanović, Nikola Jevtić, Bojan Rašković, Vukadin Milankov, Zoran Milosević, Srđan S. Ninković, Nachiappan Chockalingam, Borislav Obradović, Patrik Drid

**Affiliations:** 1Faculty of Sports and Physical Education, University of Novi Sad, 21000 Novi Sad, Serbia; 2Scolio Centar, 21000 Novi Sad, Serbia; 3Faculty of Medicine, University of Novi Sad, 21000 Novi Sad, Serbia; 4Institute for Children and Youth Health Care of Vojvodina, 21000 Novi Sad, Serbia; 5Department of Orthopedic Surgery and Traumatology, Clinical Center of Vojvodina, 21000 Novi Sad, Serbia; 6Centre for Biomechanics and Rehabilitation Technologies, Staffordshire University, Stoke on Trent ST4 2DF, UK

**Keywords:** Schroth exercise, Cobb angle, angle of trunk rotation, quality of life, posture, rehabilitation, therapeutic exercise

## Abstract

(1) Background: Idiopathic scoliosis can be defined as a complex three-dimensional deformity of the spine and trunk, which occurs in basically healthy children. Schroth scoliosis-specific exercises have shown good results in reducing idiopathic scoliosis progression. This study aimed to critically evaluate the effect size of Schroth’s method through a systematic review and meta-analysis. (2) Methods: Four databases were included in the search: PubMed, Cochrane Library, Web of Science, and Google Scholar. The following keywords were used: “Schroth exercise”, “idiopathic scoliosis”, “Cobb angle”, “angle of trunk rotation”, and “quality of life”. Only articles written in English that met the following criteria were included in our study: subjects who had idiopathic scoliosis, the Schroth method was applied, and Cobb angle or angle of trunk rotation or quality of life as outcomes. (3) Results: Ten randomized controlled trials were included in this study. The effect size of the Schroth exercise ranged from almost moderate to large, for the outcomes used: Cobb angle (ES = −0.492, *p* ˂ 0.005); ATR (ES = −0.471, *p* = 0.013); QoL (ES = 1.087, *p ˂* 0.001). (4) Conclusions: The current meta-analysis indicates that the Schroth method has a positive effect on subjects with idiopathic scoliosis.

## 1. Introduction

Idiopathic scoliosis (IS) is a three-dimensional deviation of the spinal axis and results in the appearance of frontal curves, fixed vertebral rotations, and smoothing of sagittal physiological curves, and can cause reduced spinal movement, muscle weakness near the spine, decreased pulmonary function, respiratory dysfunction, chronic pain, and psychological suffering [1,2,3,4]. It is of unknown cause and occurs mainly in healthy children [5,6]. IS is classified based on the patient’s age when it was first identified [7]. Infantile scoliosis occurs before the third year of life. Juvenile scoliosis is first detected between the ages of three and ten, while adolescent idiopathic scoliosis occurs between the age of ten and skeletal maturity [1,8,9]. In adults, scoliosis affects patients older than 18 years. Adolescent idiopathic scoliosis is the most common form of scoliosis (84–89%) [10] with a prevalence between 0.47% and 5.2% in the general adolescent population [11,12]. The three main determinants of progression are the sex of the patient, the future growth potential, and the size of the curve at the time of diagnosis [13]. An estimation of growth potential is performed by estimating the Tanner stage and Risser grade. Tanner stage 2 to 3 occurs immediately after the onset of puberty and represents the time of maximal progression of scoliosis [14]. The Risser grade (from 0 to 5) provides a useful estimate of how much skeletal growth is left by assessing the progress of the bony fusion of the iliac apophysis [7]. IS is more common in female children [15] and up to ten times more common than in male children [13]. However, in boys, there is a greater progression of curvature [16].

The Cobb angle is the most commonly used parameter for monitoring IS status because the primary goal of treatment is to stop progression and correct the deformity curves [17]. Measuring the Cobb angle is done in a standing position, mostly with X-rays [16]. The main diagnostic criterion is a Cobb angle greater than 10° [1,18]. Among other parameters for trunk deformity assessment and IS monitoring, the angle of trunk rotation (ATR) is measured with a scoliometer [19]. The subjective assessment of the patient’s condition before and after treatment is performed using standardized SRS-22 or SRS-23 (Scoliosis Research Society) questionnaires for quality of life (QoL) testing, which consists of five domains: function, pain, mental health, self-image, and satisfaction with management [20].

Physiotherapy can also be started for mild scoliosis. The most popular type of physiotherapy for scoliosis is the Schroth method. The three-dimensional Schroth method was developed a century ago in Germany by Katarina Schroth, further developed by her daughter Christa [21], and popularized in academia by her grandson, an orthopedist in private practice [22]. The program is based on sensorimotor and kinesthetic principles and includes corrective exercises, posture self-correction, breathing techniques, education, and home exercises [23,24]. Using a mirror, the patient learns to visualize their deformities and thus perform self-correction of the wrong position [25]. Tensile force, which is unique to Schroth therapy, is crucial for postural correction [23]. Numerous authorized kinesitherapy programs are used today in the treatment of scoliosis: Mischell exercises, Klapp exercises, active self-correction, Scientific Exercise Approach to Scoliosis (SEAS), DobMed, side shift, core exercises, and the Schroth method [16,25,26,27]. According to SOSORT (Society on Orthopedic and Rehabilitation Treatment) clinical guidelines, the main goals of conservative methods of scoliosis treatment are the treatment of respiratory disorders and their reduction, the treatment of back pain syndrome, and the improvement of the patient’s appearance by improved body posture. All of this is aimed at avoiding surgical procedures [15,28,29,30].

The aim of this study is to assess the effect size (ES) of the Schroth method in the treatment of IS.

## 2. Materials and Methods

### 2.1. Study Design

In conducting this systematic review and meta-analysis, we used the Preferred Reporting Items for Systematic Reviews and Meta-Analyses (PRISMA) Statement [31].

### 2.2. Data Sources and Searches

The search strategy included and identified all useful studies that applied the Schroth method in the treatment of IS. The following databases were searched: PubMed, Cochrane Library, Web of Science, and Google Scholar. The key terms used were: “Schroth exercise”, “idiopathic scoliosis”, “Cobb angle”, “angle of trunk rotation”, and “quality of life”. The search strategy is shown in Figure 1.

### 2.3. Study Inclusion and Exclusion Criteria

The inclusion of studies was limited to the following PICOS items, as recommended by PRISMA [31]. P (population): diagnosed subjects with idiopathic scoliosis, I (intervention): the Schroth method was applied, C (comparison): control group received no treatment or received standard care or other treatment, O (outcome): the Cobb angle, in degrees, is represented, and the angle of trunk rotation is also reported in degrees or quality of life measured using validated questionnaires (e.g., 22-item or 23-item Scoliosis Research Society questionnaire), and S (study design): randomized control trials published after 2000 were included. The search was limited to articles written in English. Books, book reviews, case studies, conference publications, study protocols, pilot studies, meta-analyses, and systematic reviews were excluded from the search. In October 2021, a systematic search of four databases (PubMed, Cochrane Library, Web of Science, and Google Scholar) was performed. Two researchers performed the inclusion/exclusion of studies by consultation and consensus. Mendeley software was used to enter the citations and references.

### 2.4. Data Extraction and Assessment of Methodological Quality

Two investigators independently performed data extraction after selecting studies using the inclusion and exclusion criteria. From the studies included in the meta-analysis, the table abstracts the following variables: authors, year of publication, number of subjects, age, program type, outcomes, Cobb angle, exercise time per day, program duration, and sessions per week. A study by Kuru et al. [32] for Cobb angle and QoL outcomes presented the results of the final measurement as median (min–max).

The methodological quality of each included article was assessed using the Cochrane Risk of Bias Tool, which included seven sources of bias [33]. Each included study was independently assessed by two investigators and assessed as being at high risk, unclear risk, or low risk. We used the RevMan 5.4.1 software (The Cochrane Collaboration, Copenhagen, Denmark) for the creation of risk of bias plots.

### 2.5. Data Synthesis

All statistical analyses were performed using Meta-Analyst software [34]. The effect size was estimated for the outcomes of Cobb angle, ATR, and QoL. For each study, a random effects model was used for standardized mean difference, 95% confidence interval (CI), and continuous outcomes. Cobb angle and ATR were measured by some studies in the total, and some in the thoracic and lumbar parts, which can be seen in the forest plots. If necessary, a leave-one-out analysis will be conducted if the results of individual studies lead to confusion. A value ≥ 0.2 indicates a small, ≥0.5 indicates a medium, and ≥0.8 indicates a large magnitude effect size [35]. Statistical significance was considered at *p* < 0.05. The percent heterogeneity was assessed by the Higgins I^2^ test for each outcome measured [33].

## 3. Results

### 3.1. Study Selection and Characteristics

A total of 262 studies were initially selected in four electronic databases. In total, 77 duplicates and 17 studies not written in English were immediately excluded. The remaining 168 studies were selected for further analysis. After screening the titles, 1345 studies were excluded because they did not meet the inclusion criteria. The remaining 33 studies were reviewed in full. When the full-text studies were reviewed, 24 were excluded. The remaining nine studies were included in this systematic review and meta-analysis, and one eligible study was subsequently found. The study selection process is shown in Figure 1.

Table 1 describes the characteristics of the studies we included in the meta-analysis. Two hundred and seventy-eight respondents participated in ten studies. The number of subjects ranged from 15 to 50. There were 141 subjects in the experimental Schroth group and 137 in the control group. Schreiber et al. [36] and Schreiber et al. [37] used the same cohort, but reported different outcomes in the two papers. In two studies, the control group did not receive any treatment, while in six studies, the control groups received different types of treatment, and in two studies, the subjects had a standard of care. The respondents were between 10 and 37 years old. Daily treatment ranged from 60 to 120 min, and the shortest treatment lasted 6 weeks and the longest 6 months.

### 3.2. Risk of Bias

Figure 2 and Figure 3 present the summary of the risk of bias for each included study. For the item of “random sequence generation”, all 10 studies used randomization. The concealment of allocation to the group was unclear in one study and high in five studies. The physiotherapists and participants could not be blinded to the treatment due to the nature of the intervention. For the item “blinding of outcome assessment”, four studies adopted a single-blind method to evaluate the intervention measures. Because of the objective outcome measures, the outcome data were considered low risk in 10 studies.

### 3.3. Meta-Analysis

#### 3.3.1. Cobb Angle

The Cobb angle is most often used to determine spinal deformity in patients with IS, especially for representing or diagnosing the severity of scoliotic curves. Eight studies used the Cobb angle as an outcome. After data pooling, statistical significance (ES = −0.492; 95% CI = −0.750, −0.234, *p* ˂ 0.005) and heterogeneity (I^2^ = 0%, *p* = 0.746) were shown (Figure 4). Figure 5 shows the Leave one out analysis for the Cobb angle outcome, which shows how individual studies influence the effect size.

#### 3.3.2. ATR

Four studies used ATR as an outcome. After data pooling, the statistical significance of improvement was observed as well as in the outcome of the Cobb angle, without the presence of heterogeneity (ES = −0.828; 95% CI = −1.622, −0.034, *p* = 0.041; heterogeneity: I^2^ = 80.37%, *p* ˂ 0.001) (Figure 6). Due to the large heterogeneity, the study by Mohamed and Yousef [44] was excluded from further analysis, and the results were obtained with a smaller effect size, but without the presence of heterogeneity (ES = −0.471; 95% CI = −0.842, −0.099, *p* = 0.013; heterogeneity: I^2^ = 0%, *p* = 0.471) (Figure 7).

#### 3.3.3. QoL

Three studies measured the outcome of QoL. After data pooling, when the analysis for the outcome of QoL as a difference between the final and initial measurements in the experimental and control groups was achieved, statistical significance was achieved with negligible heterogeneity (ES = 1087; 95% CI = 0.597,1.578, *p* ˂ 0.001; heterogeneity: I^2^ = 30.06%, *p* = 0.239) (Figure 8).

## 4. Discussion

In our systematic review, the effect size of the Schroth method on subjects with IS, of which there were a total of 278, was calculated by combining the results of 10 included studies, which was the aim of this study. The most important goals of physiotherapists who work with subjects who have IS are the correction and prevention of scoliotic disorders. We presented the effect size for all three outcomes that we calculated by comparing the results of post-treatment intervention minus pre-treatment intervention in the experimental and control groups. The overall effect size achieved statistical significance for the outcome of the Cobb angle (ES = −0.492, ≥0.5—almost moderate effect size) (Figure 4). The effect size achieved statistical significance for the ATR outcome as well (ES = −0.471, ≥0.5—almost moderate effect size), with the prior exclusion of one study (Figure 7). The effect size for the QoL outcome was also statistically significant (ES = 1.087, ≥0.8—large effect size) (Figure 8).

In our systematic review, respondents with IS problems were investigated. The Schroth method was used as the only type of corrective program. The respondents ranged from 10 to 37 years of age, which diminishes the relevance because older respondents were not included in finding out how this method affects them. The problem is that there are no such published studies. The study by Lee and Lee [43] involved older respondents than the other studies, and thus increased the age range of the participants. The leave-one-out analysis for the outcome of Cobb angle shows the values of the magnitude of the effect that this study was excluded from and that participants from 10 to 22 years of age participated in the analysis (ES = −0.492 vs. ES = −0.499) (Figure 5). In this population, the results realistically show the effect size for this number of included studies. We believe that there were respondents from the older population in these ten included studies, and on such a sample, the results of the overall magnitude of the effect would be less relevant. In these ten studies, we were able to estimate the effect sizes for three outcomes. If there were even more common outcomes, the results of this meta-analysis would be even better.

All studies included together had a low risk of bias. A high risk of bias was found for the item “allocation concealment”. A study by Kuru et al. [32] had a high risk of bias in the item “selective reporting” because, for the Cobb angle and QoL outcomes for the final measurement, they presented the results as the median (min–max). The pooled effect size estimate was consistent, indicating the absence of heterogeneity in the Cobb angle and ATR outcomes and the presence of negligible heterogeneity (I^2^ = 30.06%) in the QoL outcome. The results of the QoL outcome effect size (ES = 1087, ≥0.8—large effect size) show the improvement that subjects subjectively felt after applying the Schroth method. In our opinion, the total magnitude of the effect of the Schroth method application on subjects with IS is certainly within the limits of moderate magnitude, i.e., it ranged from an almost moderate effect to a large magnitude of the effect for different outcomes.

Our search included multiple electronic databases and additional resources and was comprehensive. We acknowledge that other studies written in other languages have been neglected. The study by Kim and Park [41] used two groups that used the Schroth method, while one of them also used additional respiratory muscle exercises using SpiroTiger (Idiag, Fehraltorf, Switzerland). We decided by consensus that the group that used additional respiratory muscle exercises should be the control group. This can be a potential bias in the review process, as it certainly reduces the effect size in the Cobb angle outcome. This can be seen in Figure 4, where the effect size of this study is the only one with a positive sign (ES = 0.102), which means that, in this case, the results of the control group are favored over those of the experimental group. Therefore, a sensitive leave-one-out analysis for the outcome of the Cobb angle was performed, which shows results that exclude one study. The results (ES = −0.533 vs. ES = −0.492) show that the effect size would be entirely within the values for the moderate effect size (Figure 5). A study by Duangkeaw et al. [38] used two groups that used Schroth exercises, while one of them also used Kinesio tape. As in the previous case, as an experimental group, we used the one that only applied the Schroth exercises. The results did not favor the control group, so we did not do a leave-one-out analysis for the ATR outcome as in the previous case. Potential bias in the review process may include decisions made in connection with the study by Kuru et al. [32]. In the previous paragraph, we discussed the way in which this study presented the results. In order not to exclude it from further analysis, we decided to use the median values as the mean value, and imputed the SD values from the values in the initial measurement. These decisions are in line with the recommendations on imputation and missing data provided by the studies of Furukawa et al. [45], Hozo et al. [46], and Higgins et al. [47]. This study gives results after 6 weeks, after 12 weeks, and after 24 weeks, but because the results are presented in the same, already mentioned way, we did not include them in the analysis because there would be too many imputations. This same study used another control group that used a home exercise, but we did not include it in our analysis. Studies by Schreiber et al. [36] and Schreiber et al. [37] used the same cohort, but different outcomes, and we included both in the analysis. In this case, the real number of subjects included in our study decreased from 278 subjects to 228. Based on the number of studies included and the common outcomes they measured, we were unable to perform any subgroup analysis, nor by the duration of treatment, by age, or based on the magnitude of scoliotic changes, gender, or according to the Tanner stage or Risser grade. Two hundred and fifty-five female respondents and 23 male respondents were included. Only three studies used subjects of one sex, and therefore no subgroup analysis could be performed.

The results of the meta-analysis conducted by Park et al. [48] show substantial heterogeneity (I^2^ = 75.67%) among the included studies, while the magnitude of the effect was almost large (ES = 0.784, ≥0.8—almost large effect size). This is the magnitude of the effect based on all outcomes for all studies, which, in our opinion, has led to this heterogeneity. The effect size for the Cobb angle outcome was moderate (ES = −0.65, ≥0.5—moderate effect size), with no report of heterogeneity. The effect size for ATR outcome was moderate (ES = 0.53, ≥0.5—moderate effect size), with no report of heterogeneity. The effect size for the QoL outcome was large (ES = 0.76, ≥ 0.8—large effect size), also without reports of heterogeneity. The differences in the size of the effect between our study and the study of Park et al. (48), according to us, are in the degrees of freedom that were used in the outcomes. For the Cobb angle outcome, we had 8 degrees of freedom (df = 8) and they had 10 (df = 10). For the ATR outcome, we had 4 (df = 4) and they had 6 (df = 6), while for the QoL outcome, both studies had 2 (df = 2). In the last outcome, our assumption is not confirmed because there were the same number of degrees of freedom. The differences in the results may also come from the fact that in the study by Park et al. (2017), out of 15 included studies, seven did not have a control group, which further complicates the analysis and reduces the relevance of the results. A study by Burger et al. [49] included four studies. A meta-analysis was only performed for the outcome of QoL (ES = 0.83, ≥0.8—large effect size), without heterogeneity (I^2^ = 0%), (df = 1), while for the Cobb angle, significant heterogeneity was also recorded, and the results are not even shown. No meta-analyses were performed in other systematic reviews dealing with the Schroth method, so our results can only be compared with the study of Park et al. [48] so that we can draw conclusions about the impact of Schroth exercise on subjects with IS. The results of meta-analyses that dealt with the problems of kyphosis and lordosis using different corrective programs [50,51] report a moderate effect size, so the results of this study encourage our subjects to use Schroth exercises in therapy. In our analysis, all included studies were RCTs. We combined the results of several studies and calculated the overall effect size and did not deal with the results of individual studies.

The limits of this systematic review and meta-analysis are as follows. First, only articles written in English were included in the review. Second, the number of articles found is still small, despite a thorough search, with a small number of common outcomes being assessed. Third, the studies included in our analysis had a relatively small number of respondents ranging from 15 to 50. Fourth, the disadvantage of the studies included was that they did not yet have the common outcomes they measured.

## 5. Conclusions

Our meta-analysis indicates a positive effect size of the Schroth method on patients with IS. The effect size ranged from almost moderate to large depending on the outcome we analyzed. From this comprehensive analysis, we could conclude that future research should include an even larger number of studies (as currently, only so many are available), there should be a larger number of respondents, and that a larger number of common outcomes would be needed to assess of the effect size. Moreover, good comparators (control groups) should be set up and all conditions met to avoid the risk of bias during the research process. Therefore, there is an incentive to deal with the effects of the Schroth method on patients with IS in the future in order to obtain the best possible results. We believe that our study can be of benefit to all those dealing with IS problems, especially clinicians, physiotherapists, and physical activity specialists, and will encourage future scientific research.

## Figures and Tables

**Figure 1 ijerph-19-16730-f001:**
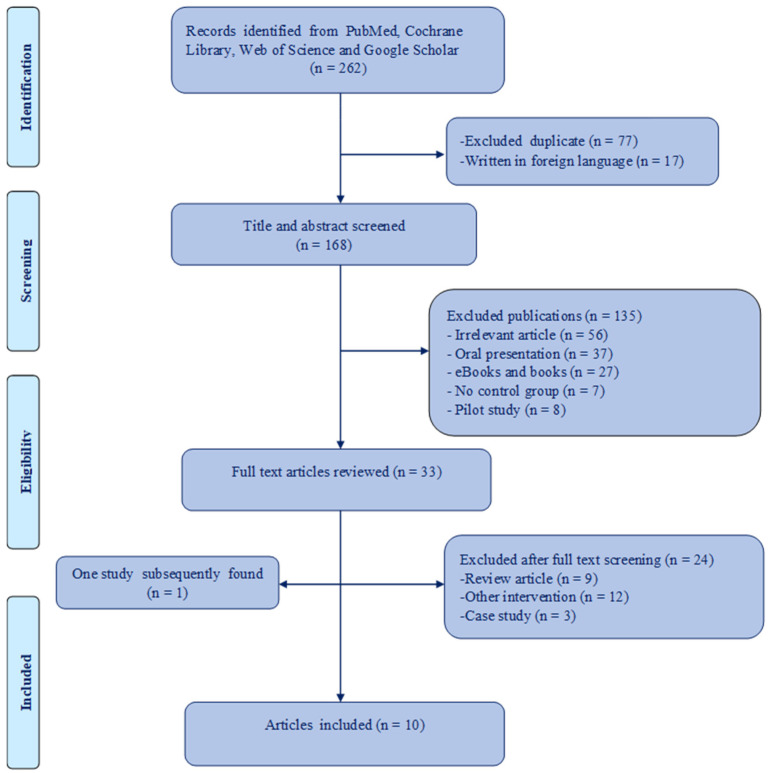
Flow chart of search.

**Figure 2 ijerph-19-16730-f002:**
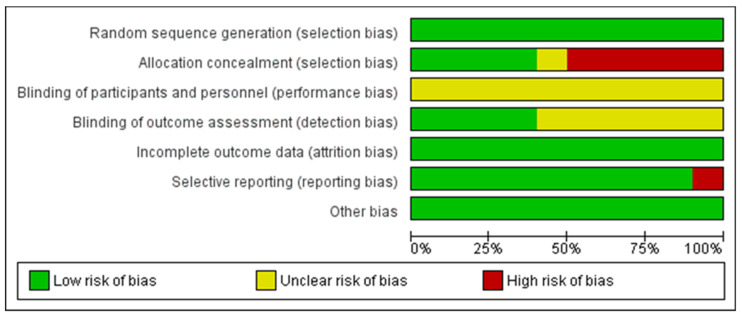
Risk of bias graph: authors’ assessment of each item expressed as a percentage for all included studies.

**Figure 3 ijerph-19-16730-f003:**
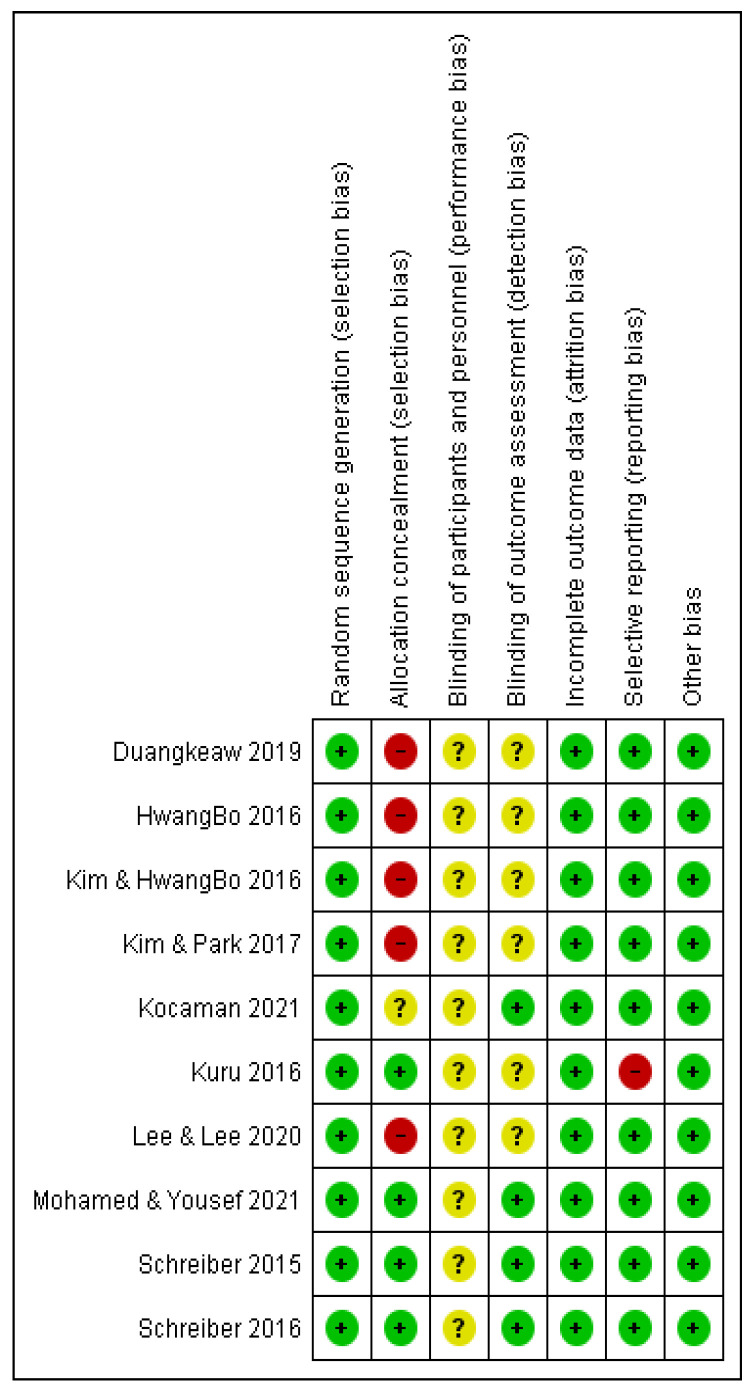
Summary of risk of bias: review of the authors’ judgments about each item for each included study. Green (+)—low risk, yellow (?)—unclear risk, red (-)—high risk.

**Figure 4 ijerph-19-16730-f004:**
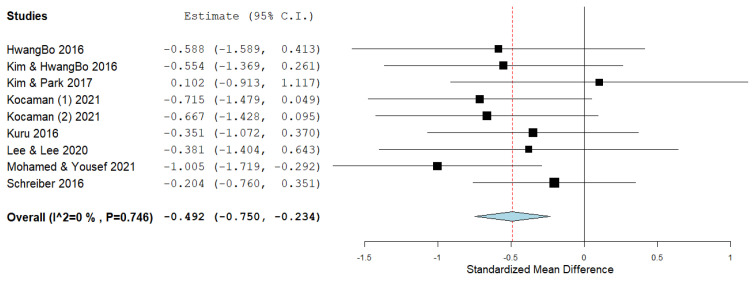
Effect size for the outcome Cobb angle.

**Figure 5 ijerph-19-16730-f005:**
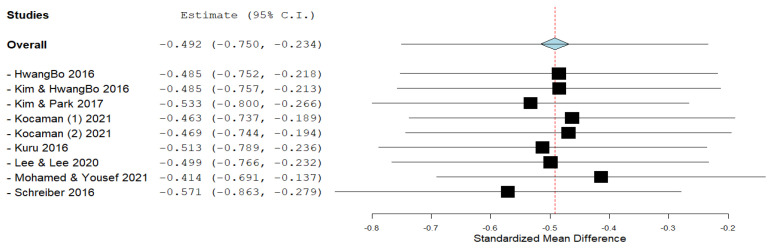
Leave-one-out analysis for the outcome Cobb angle.

**Figure 6 ijerph-19-16730-f006:**
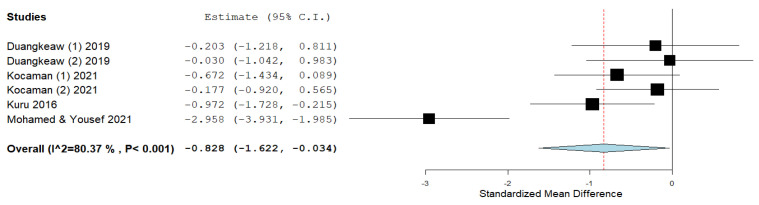
Effect size for the outcome ATR.

**Figure 7 ijerph-19-16730-f007:**
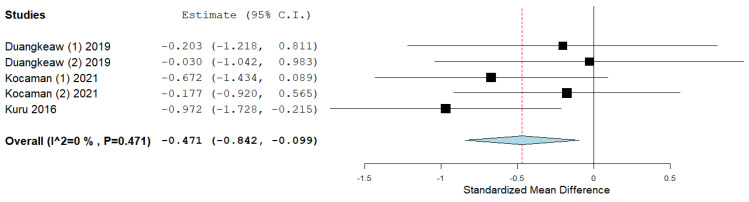
Effect size for the outcome ATR, without the study by Mohamed and Yousef [44].

**Figure 8 ijerph-19-16730-f008:**
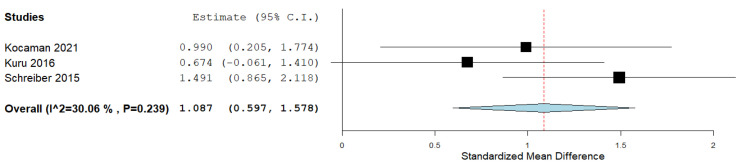
Effect size for the outcome QoL.

**Table 1 ijerph-19-16730-t001:** Description of the studies.

Study	*N*	Program Type	Outcome	Cobb Angle	Age	Exercise Time	Duration	Sessions
						Per Day		Per Week
Duangkeaw 2019 [38]	16	Schroth 3D exercise	ATR	N/A	10–18	120 min	6 weeks	2
		vs.	Inspiratory muscles strength					
		Kinesio tape with Schroth	Expiratory muscles strength					
		Exercise	Muscle endurance of back					
			General mobility					
HvangBo 2016 [39]	16	Schroth exercise	Cobb angle	SEG. 22.07 ± 6.81	18.14 ± 1.6	N/A	12 weeks	3
		vs. Pilates exercise	Psychological factors	PEG. 21.20 ± 3.95	18.88 ± 1.55			
Kim & HvangBo 2016 [40]	24	Schroth exercise	Cobb angle	SEG. 23.63 ± 1.5	15.6 ± 1.1	60 min	12 weeks	3
		vs. Pilates exercise	Weight distribution	PEG. 24.0 ± 2.6	15.3 ± 0.8			
Kim & Park 2017 [41]	15	SERME	Cobb angle	SER. 24.49 ± 8.32	17.75 ± 4.71	60 min	8 weeks	3
		Schroth 3D exercise	Pulmonary function	SEG. 27.16 ± 12.44	15.57 ± 2.70			
			Functional movement screen					
Kuru 2016 [32]	30	Schroth 3D exercise	Cobb angle	10–60°	10–18	90 min	6 weeks	3
		vs. control (no treatment)	ATR				3 months	
			Asymmetry				6 months	
			Quality of life					
Kocaman 2021 [42]	28	Schroth exercise	Cobb angle	10–30°	10–18	60 min	8 weeks	3
		vs. Core exercise	ATR					
			Cosmetic trunk deformity					
			Spinal mobility					
			QoL					
Lee & Lee 2020 [43]	15	Schroth exercise	Cob angle	SEG. 22.11 ± 7.58	18.88 ± 3.06	120 min	12 weeks	2
		vs. control (no treatment)	VTR	Con. 22.17 ± 7.27	24.14 ± 12.69			
			DV					
			Correction rate					
Mohamed & Yousef 2021 [44]	34	Schroth exercise	Cobb angle	SEG. 20.42 ± 2.57	14.50 ± 1.20	60 min	6 months	3
		vs. PNF	ATR	PNF. 20.21 ± 2.80	14.90 ± 1.40			
			Total static plantar pressure					
			6MWT					
Schreiber 2015 [36]	50	Schroth exercise	Quality of life	10–45°	10–18	60 min	3 months	5
		Standard of care	Back extensor strength				6 months	
Schreiber 2016 [37]	50	Schroth exercise	Cobb angle	10–45°	10–18	60 min	6 months	5
		Standard of care	Sum of curves					

N: number of subjects in the group; ATR: angle of trunk rotation; SEG: Schroth exercise group; PEG: Pilates exercise group; Con: control group; SER: Schroth 3D exercise in combination with respiratory muscle exercise; VTR: vertebral rotation angle; DV: difference in ratio volume; PNF: proprioceptive neuromuscular facilitation; 6MWT: six-minute walk test; N/A: not applicable.

## Data Availability

Not applicable.

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
