# Peer review of "Application of the Schroth Method in the Treatment of Idiopathic Scoliosis: A Systematic Review and Meta-Analysis"

_ijerph, 2022, doi:10.3390/ijerph192416730_

Round 1
Reviewer 1 Report
In this study, Dimitrijevic et.al evaluated the effectiveness of the Schroth method in the treatment of idiopathic scoliosis by systematic review and meta-analysis. The study objective was clearly stated, and the literature search was comprehensively conducted. After analyzing 10 related clinical trials involving 278 patients, the authors came to the conclusion that the Schroth method has a positive effect on subjects with idiopathic scoliosis. This meta-analysis study was well performed, and the conclusion could be useful for the clinicians. The only comment I have is that the authors might want to check the degree of scoliosis and the treatment duration in the trials to see whether the two factors could affect the treatment efficiency.
Author Response
First of all, thank you for all your suggestions.
Please see the attachment for our response to all comments.

Reviewer 2 Report
This systemic review/meta-analysis summarized 10 eligible studies's results on the “Schroth method in the treatment of idiopathic scoliosis”. Overall, the meta-analysis is sound and the writing is good with minor English Issues.
Following are some minor issues or comments that needs to be addressed.
1. Table 1: Column “Sessions per week”, authors put 2X or 3X week for different studies, it is confusing. I think put just number of sessions is sufficient and clearer.
2. Figures 2-7: Text in the figures are in gray color and font size is too small to read. They should be black color and main manuscript text.
3. Lines 221-222, “This number of “ is not clear what to indicate. Does the author indicated the 10 studies included in this systemic review and meta-analysis?
4. Line 227, “A study Kuru et al” should be “A study by Kuru et al”.
5. The authors said the age affected the effect size, should the author separate different age of IS and evaluate effect size so that to provide guidance to readers interested in this method for treatment their patients. It is common sense the earlier the correction method is applied; the higher and better correction outcome will achieve. But it is still useful to evaluate how this method’s effective size for older patients (adult).
6. Line 282-284 “we had 8 degrees 282 of freedom (df = 8) and they (df = 10)” …. What “they” mean here?
7. Line 286,” In the study Park et al”, should be “In the study by Park et al”.
Author Response

(The authors gave the same response as above.)
